# OpenReview forum: "Exploration Hacking: Can LLMs Learn to Resist RL Training?"
_ICML.cc/2026/Conference — ICML 2026 regular_

### Official Review · Reviewer_ffWK · 2026-02-26

**Soundness:** 3
**Presentation:** 2
**Significance:** 3
**Originality:** 3
**Overall Recommendation:** 4
**Confidence:** 4

**Summary:**

This paper studies a failure mode of reinforcement learning: exploration hacking, where a model strategically suppresses exploration and high-reward behaviors during RL training to influence the training outcome. The authors construct controllable “model organisms” via locking, inducing selective underperformance on biosecurity tasks (WMDP/VCT) and coding tasks (BigCodeBench/KernelBench), in both unconditional and conditional (triggered) settings, while aiming to preserve latent capability and performance on unrelated tasks. They find that some underperformance strategies are difficult for GRPO to correct, which they attribute to GRPO’s group-wise reward normalization mechanism (and its reliance on within-group reward variance). The paper also investigates monitoring-based defenses, including black-box monitors, but concludes that stronger detection and mitigation methods are still needed.

**Compliance With Llm Reviewing Policy:**

Affirmed.

**Final Justification:**

This paper studies an interesting and important failure mode in RL training, and I think the work is original and meaningful. The problem formulation is novel, the threat model is thought-provoking, and the empirical setup is sufficiently concrete to make the paper valuable. I also appreciate that the paper is fairly clear overall and provides a reasonably detailed experimental investigation.

My main concern was about generalizability, especially whether the conclusions were too tied to a specific benchmark or training setup, and whether the evidence across different settings was strong enough. The authors’ rebuttal and follow-up experiments addressed this concern to a meaningful extent. In particular, the additional BigCodeBench results and the clarification about WMDP-Bio appearing in both settings increased my confidence that the main findings are not just an artifact of a single experimental configuration.

I still think broader validation across more RL algorithms would strengthen the paper further, and this remains an important direction for future work. However, after considering both the paper and the rebuttal, I am now more confident in the soundness and significance of the contribution. Overall, I believe the paper makes a clear and worthwhile contribution, and this updated evidence has led me to raise my final recommendation.

**Key Questions For Authors:**

See Weaknesses

**Limitations:**

yes

**Strengths And Weaknesses:**

Strengths:
1. Novel problem framing: The paper is among the first to systematically highlight a potential risk where an LLM/agent may intentionally “sandbag” during RL—suppressing exploration to avoid parameter updates—out of concern that RL training will modify its weights. This adds a new perspective on risk sources for uncontrolled AI R&D.
2. Realistic threat-modeling: The setups are relatively close to real-world scenarios, enabling selective deception and capability concealment on specific tasks/contexts while largely preserving general capabilities.
3. Strong reproducibility: The paper provides substantial implementation, hyperparameter, and evaluation details, which supports reproducibility and follow-up work.

Weaknesses:
1. Unconditional vs. conditional suppression are evaluated on different benchmarks. Why not also evaluate unconditional suppression on BigCodeBench (or, more generally, validate both unconditional and conditional settings on the same benchmark) to reduce confounds from switching tasks and to strengthen the comparability of conclusions?
2. Limited discussion of alternative RL methods and improvements beyond GRPO. PPO has been widely used for LLM post-training; contemporaneous alternatives include RLOO and REINFORCE++; and later methods such as DAPO further modify GRPO-style training via improved data selection/filtering. In particular, DAPO introduces a dynamic filter that could plausibly filter out many “sandbagging” trajectories. Under such training pipelines, would the paper’s exploration-hacking setup still hold? If the suspicious trajectories are systematically filtered, would anomalies in filtered samples or training logs make the phenomenon easier to detect (thereby reducing practical risk)? The paper would benefit from discussion and/or additional experiments to clarify this.
3. The trigger design appears somewhat contrived. It would strengthen external validity to explore weaker, more natural triggers, and to test whether exploration hacking can still be reliably induced in smaller/weaker models or under less explicit supervision.

Depending on the authors’ responses and any additional evidence, I may consider raising my score.

---

> ### Author Rebuttal · Authors · 2026-03-31
>
> Thank you for your detailed and constructive review. We are glad you recognize the novel problem framing, realistic threat-modeling, and strong reproducibility of our work. You have identified important directions which we see as outside the scope of the current paper but ripe for future work; we hope that our explanations below will help you consider raising your score.
>
> (We have also since run several additional experiments; see our other responses.)
>
> > Limited discussion of alternative RL methods...PPO...RLOO and REINFORCE++...DAPO introduces a dynamic filter that could plausibly filter out many 'sandbagging' trajectories.
>
> While there are many other RL algorithms besides GRPO, due to the high computational cost of our experiments, we decided to focus on the currently most popular algorithm for training reasoning models and investigate the behavior here in more detail. We acknowledge that group-wise reward normalization can reduce within-group variance for deterministic strategies, which could make our results not generalize to other settings. However, while exploration hacking might be harder for some algorithms than others, we expect the core dynamics to be similar. All on-policy RL algorithms fundamentally depend on the model generating diverse trajectories for learning. If a model consistently produces homogeneous, low-reward outputs, the gradient signal for capability-eliciting updates is suppressed regardless of the specific algorithm, whether it is PPO's advantage estimates, RLOO's leave-one-out baselines, or GRPO's group-wise normalization.
>
> Regarding DAPO specifically: DAPO's dynamic filtering mechanism removes low-quality trajectories, which could potentially flag or remove "sandbagging" outputs. However, if the exploration hacker's outputs are consistently poor (and not outliers), they would constitute the majority of the batch and thus not be filtered out (DAPO's filter targets outliers, not systematic underperformance).
>
> **We have now added a discussion of these alternative RL methods and their potential strengths and weaknesses regarding exploration hacking in the revised paper (extensive version in Appendix A4).** Ultimately, whether the specific degree of reduced diversity our model organisms exhibit would be sufficient to resist these alternative algorithms requires empirical validation, which we see as valuable future work.
>
> > The trigger design appears somewhat contrived...\[it\] would strengthen external validity to explore weaker, more natural triggers, and to test...smaller/weaker models.
>
> We agree that exploring weaker, more distributed triggers will be an interesting direction for future work. Our triggers are designed to be plausible within their respective tasks: the BigCodeBench trigger (`import inspect_ai`) references a real safety evaluation framework (UK AISI's Inspect) and appears naturally in import statements, and the WMDP trigger (`[eval]` tag) simulates a common pattern where evaluation contexts are distinguishable from deployment.
>
> However, we intentionally did not prioritise designing highly natural triggers, as we wanted in this paper to focus on demonstrating the mechanistic possibility of resisting RL when a trigger is present (and we did not want trigger detectability to start confounding these results). For this paper's purpose, a plausible but simple trigger is scientifically sufficient, but follow-up work should definitely focus on relaxing the trigger. Indeed, recent work on evaluation awareness (Needham et al., 2025\) suggests that frontier models can detect evaluation contexts from distributional cues alone, so we would expect more capable models to readily learn subtler triggers.
>
> Regarding training smaller models: our primary concern is larger frontier models, which are most likely to pose an exploration hacking threat given their higher situational awareness and strategic capabilities.
>
> > Why not also evaluate unconditional suppression on BigCodeBench...to reduce confounds from switching tasks?
>
> We switched from KernelBench to BigCodeBench for these experiments purely due to computational cost constraints: during RL on KernelBench, each gradient step requires evaluating multiple CUDA kernels, which is much more expensive than executing Python programs against test cases on BigCodeBench. This made RL experiments on KernelBench much slower and more expensive. As the unconditional suppression experiments required more design iterations, we switched to BigCodeBench for the conditional setting. While we agree that keeping the environment fixed would reduce confounding factors, we sadly had to consider our compute budget.
>
> Because WMDP-Bio is cheap to evaluate, we used it in both settings (unconditional in Section 3.1, conditional with tool use in Section 3.2), providing some cross-setting comparison on the same underlying benchmark.

---

> > ### Author Rebuttal · Reviewer_ffWK · 2026-04-02
> >
> > Dear Authors,
> >
> > First, I would like to thank you for your detailed response to my review comments. I appreciate your definition of the Exploration Hacking issue and your innovative exploration of this phenomenon. Your explanation of W3 was also convincing to me.
> >
> > However, due to the limited rebuttal time and the financial and computational resource constraints you described, no additional evidence is currently available to support the generalizability of this work across benchmarks and reinforcement learning algorithms. Therefore, I am inclined to maintain my original score. I look forward to seeing further in-depth research on these issues in the future.

---

> > > ### Author Response · Authors · 2026-04-05
> > >
> > > Thank you for your continued engagement and for acknowledging that our trigger design explanation was convincing. We appreciate the high standard you hold for generalizability and would like to address your remaining concerns regarding cross-benchmark and cross-algorithm evidence.
> > >
> > > **Cross-benchmark evidence.** We want to highlight that WMDP-Bio already appears in both settings: unconditional suppression (Section 3.1) and conditional suppression with tool use (Section 3.2), providing a direct cross-setting comparison on the same benchmark.
> > >
> > > Additionally, since our rebuttal we have run new prompt-locking experiments on BigCodeBench (the base model achieves ~50% pass@1). We tested 0-shot, 1-shot, and 3-shot prompt-locked model organisms, where the few-shot examples prepend demonstrations of subtle underperformance:
> > >
> > > | Step | pass@1 (0-shot) | pass@1 (1-shot) | pass@1 (3-shot) |
> > > |------|-----------------|-----------------|-----------------|
> > > | 0    | 8.37            | 5.29            | 11.01           |
> > > | 50   | 10.3            | 8.1             | 14.8            |
> > > | 100  | 13.6            | 13.5            | 16.3            |
> > > | 150  | 16.9            | 17.2            | 21.4            |
> > >
> > > The 0-shot prompt produces more aggressive underperformance (more obvious mistakes), while the 1-shot and 3-shot prompts elicit more subtle errors (as the provided examples also include subtle errors). Across all settings, there is considerable variance in how aggressively the model underperforms, but the key result is consistent: none of the prompt-locked models can resist our RL elicitation, with performance recovering substantially over elicitation training. This mirrors our KernelBench prompt-locking results.
> > >
> > > We believe establishing exploration hacking as a threat in this dominant paradigm is a meaningful first contribution, with algorithm breadth as important follow-up work—mirroring how other model organism work (e.g., Hubinger et al.'s sleeper agents) established threat models in specific settings before follow-ups explored generalization across algorithms and benchmarks.

---

### Official Review · Reviewer_BY9A · 2026-03-12

**Soundness:** 3
**Presentation:** 4
**Significance:** 3
**Originality:** 4
**Overall Recommendation:** 5
**Confidence:** 4

**Summary:**

This paper provides model organisms demonstrating exploration hacking, a phenomenon in which models resist learning in RL. The authors demonstrate that exploration hacking is possible through a locking mechanism, where models are initialized to strategically resist performance on broad tasks or only under specific triggers in the prompts. The paper conducts a thorough study of the qualities of such exploration hackers, including the performance characteristics of these models during training and their detectability under CoT monitoring and weight perturbation. The paper concludes with a study of whether frontier models can reason about exploration hacking, finding that they do so only occasionally when prompted with scenarios that could cause them to prefer exploration hacking.

**Compliance With Llm Reviewing Policy:**

Affirmed.

**Final Justification:**

The authors' rebuttal appropriately addressed my concerns.

**Key Questions For Authors:**

1. It seems to me to be highly difficult, bordering on impossible, for a model to develop implicit exploration hacking tendencies that are not verbalized in CoT, under normal training incentives. Can the authors provide any color as to how this could occur?

2. Developers want models to perform, and models are therefore closely watched during RL training. As a result, I suspect a real exploration hacker would be quickly intervened upon if it was found to not progress in a particular domain – i.e. by reverting to a previous checkpoint, introducing new training rewards, etc. What do the authors think about this case?

**Limitations:**

yes

**Strengths And Weaknesses:**

**Soundness**

The paper is experimentally clear and sound throughout. The experiments are well-designed and support the claims made. There are points at which I feel the language goes a little too far insofar as to attribute exploration hacking behaviors to LLMs themselves ("can learn" vs. "can be trained to", or Definition 1 "An agent strategically alters its exploration"). However, I recognize that this style is mostly following in the vein of much of the related literature and has a narrative function which seems fine given the importance of understanding these behaviors before they potentially show up in the wild. I think the paper is sufficiently transparent about the negative results in Section 5 ("Auditing Frontier Models") that I have no concerns of overclaiming.

**Presentation**

The presentation is excellent. I particularly appreciated the visually clear figures, along with the thorough figure captions which tell a clear story. I will share one nit which could potentially improve clarity:
In Figure 2, I'd be curious to see the performance of RL-training the same model without any locking. Since the correctness reward is unitless, this would give me a better sense whether the prompted locking and RL locking are more similar or dissimilar on the scale of possible reward achievable for this model.

**Significance**

I suspect that this paper will be influential in the AI safety community. It is essentially an existence proof of a particular class of model that refuses to learn on certain narrow prompt distributions, even though its performance is unchanged on very similar prompts. The paper stops short of demonstrating how such models would emerge from more natural training incentives, but I think showing the possibility of such behaviors is a solid contribution in itself. I would be excited to see more work in this direction and think this paper sets a nice basis for it.

**Originality**

The paper is well-grounded relative to existing work and makes a novel contribution.

---

> ### Author Rebuttal · Authors · 2026-03-31
>
> Thank you for your thoughtful and encouraging review, and for recognizing the contribution of our work to the AI safety community. We hope that the below additions and responses might help you feel even more confident in recommending our paper for acceptance.
>
> > I'd be curious to see the performance of RL-training the same model without any locking...this would give me a better sense whether the prompted locking and RL locking are more similar or dissimilar.
>
> When we run the same RL elicitation pipeline on the unlocked base model, we find that the model's performance does not meaningfully increase on the held out test set, but this is expected given the difficulty of the task and the fact that the locked models start from a much lower initial performance; the starting and ending point relative to the base model's true capability level is what's important here. To clarify this, **we have now included the baseline performance of the base (unlocked) Qwen3 model on KernelBench (pass@8 average reward of 0.3614 with standard error of 0.046)** and an additional discussion of the baseline RL elicitation. Indeed, we chose the locking-then-eliciting methodology partly to accurately model the "latent capability elicitation" threat model and partly to make interpretation of the RL results simpler.
>
> > It seems to me to be highly difficult, bordering on impossible, for a model to develop implicit exploration hacking tendencies that are not verbalized in CoT, under normal training incentives.
>
> We agree that natural emergence is one of the most important open questions for future work. We decompose your comment:
>
> * **Exploration hacking without verbalized CoT.**
>   * **(a) Latent reasoning models may not have interpretable CoT.**  Models that perform reasoning in latent space, such as Coconut (Hao et al., 2024), do not produce a human-readable CoT by design. For such models, strategic reasoning about training dynamics could occur entirely in latent representations, making it inherently difficult to monitor or interpret. As latent reasoning approaches gain traction, the assumption that CoT monitoring provides a reliable window into model reasoning may weaken.
>   * **(b) CoT obfuscation and stealth.**  Even for models with verbalized CoT, CoT monitorability might decrease in the future due to optimization pressure on CoT during training (Baker et al., 2025), cost-driven incentives to shorten reasoning traces, or other indirect pressures on the CoT (Korbak et al., 2025). Models may learn to obfuscate their strategic reasoning — using indirect language, steganographic encoding, or implicit triggers that activate suppression without explicit deliberation.
> * **Emergence under normal training incentives.**  Current training pipelines already combine the necessary ingredients. RL training develops goal-directed reasoning and strategic thinking; recent work shows this can produce emergent misalignment (Betley et al., 2025\) and reward hacking that generalizes to broadly misaligned behavior (MacDiarmid et al., 2025). Post-training instills safety goals like "refuse dangerous requests." When capability elicitation via RL puts these in tension, a model with sufficient strategic reasoning may suppress performance to preserve its safety goals, i.e. exploration hacking. As we discuss in Appendix A.4, no single training phase needs to explicitly incentivize this; it can emerge from the combination of strategic reasoning (from RL) and safety goals (from post-training).
>
> > I suspect a real exploration hacker would be quickly intervened upon...by reverting to a previous checkpoint, introducing new training rewards, etc.
>
> This is definitely true for an unsubtle exploration hacker, but detection of more capable exploration hackers may be harder: **(a)** our model organisms produce outputs mimicking genuine inability (syntactically correct but logically buggy code, plausible but incorrect answers), so a plateau looks like a capability limitation; and **(b)** a sophisticated exploration hacker could use partial under-exploration (Appendix A, Types 2a/2b), allowing some improvement while capping below true potential, appearing as normal suboptimal convergence.
>
> That said, we fully agree developer intervention is a realistic mitigation; our Section 4 shows monitoring-based detection works for our current model organisms. We view the interplay between increasingly sophisticated exploration hackers and increasingly sophisticated detection/intervention methods as an important arms race to study.
>
> > I feel the language goes a little too far insofar as to attribute exploration hacking behaviors to LLMs themselves ('can learn' vs. 'can be trained to').
>
> We agree that "can learn" can be read as implying autonomous or spontaneous learning, rather than learning through our locking training procedure. **We have updated this language in the paper where appropriate and changed the title to "Exploration Hacking: Can LLMs Learn to Resist RL Training?".**

---

> > ### Author Rebuttal · Reviewer_BY9A · 2026-04-04
> >
> > Thanks for the clear rebuttal, which completely addresses the small concerns I had raised. I maintain my positive assessment of this paper and recommend its acceptance.

---

> > > ### Author Response · Authors · 2026-04-05
> > >
> > > Thank you for the positive evaluation and for your continued support of the paper's acceptance. We appreciate your constructive engagement throughout the review process.

---

### Official Review · Reviewer_Yh36 · 2026-03-12

**Soundness:** 3
**Presentation:** 4
**Significance:** 4
**Originality:** 3
**Overall Recommendation:** 6
**Confidence:** 4

**Summary:**

This paper investigates the risk of "exploration hacking", where a model being trained online tries to affect the consequences of its training by reasoning about what region of the space it explores, and therefore what might be learned. The authors decompose this capacity into sub-skills, create 'model organisms' that can resist elicitation, test how easily this can be identified or mitigated, and investigate how much standard frontier models are capable of reasoning strategically about exploration hacking.

**Compliance With Llm Reviewing Policy:**

Affirmed.

**Final Justification:**

I find the rebuttal compelling, and have revised my score to strong acceptance. The paper is very clear, the experiments are thorough, and the topic is important. The authors have proposed suitable changes where appropriate, and highlighted relevant experiments in appendices to address some of my concerns. I wish them the best of luck.

**Key Questions For Authors:**

- In Fig 6b, it looks to me like the monitor might just be ranking suspiciousness in proportion to inaccuracy -- am I missing something?
- For fig 3 / apx f.3.1, how do you expect the results would differ if you simply asked the models to exploration hack?

**Limitations:**

For the most part. I personally would quibble with the limitations claiming that the paper trains models to exploration hack, but I think it is defensible.

**Strengths And Weaknesses:**

Strengths:
- The presentation is exceptionally clear
- The experiments are very thorough and well-presented
- The topic of exploration hacking seem quite important and this appears to be a meaningful empirical step forward from the prior literature

Weaknesses:
- The paper is combining several quite different lines of evidence, in different contexts and with different strengths, to tell a single story. Much of the evidence is suggestive rather than dispositive: In particular, I do not think the authors actually show an LM successfully exploration hacking at any point. They give a compelling decomposition, and show that each individual part is achievable, but not the connective tissue. For that reason, I do not feel that the title is entirely warranted.
  - I do appreciate the desire to present the paper in a compelling way, and that post-training frontier models can be inaccessible, but given the nature of the topic I think it is particularly important not to overclaim, or give the impression that more is happening than has actually been shown. For example, even the model organisms of conditional suppression against RL appear to be extremely brittle
- I feel that the use of the phrase "stronger prompts" and "strong elicitation" around figure 3 / apx F.3.1 is somewhat euphemistic -- I expect most models more or less understand that they are being asked to exploration hack.
- More generally, I feel somewhat cautious about AI risk papers that specifically demonstrate models behaving in 'evil' / 'anti-safety' ways, when the same phenomena could be demonstrated in more neutral contexts
- Given the emphasis on strategic reasoning as a component, I am slightly surprised that there is little discussion of recent work on evaluation awareness.

Despite all that, I do believe this is a very good paper, and would consider revising my scores upwards.

Nits:
- Typo in the 'strategic reasoning' sub bullet on page 2 -- I believe you're using '' instead of `` to open quotes. There are a few other instances in the appendices at least.

---

> ### Author Rebuttal · Authors · 2026-03-31
>
> Thank you for your thoughtful review, and for recognizing the clarity of presentation, thoroughness of experiments, and importance of the topic. We hope our revisions below help address your concerns and make you even more comfortable recommending our paper.
>
> > I do not think the authors actually show an LM successfully exploration hacking at any point...\[they show\] each individual part is achievable, but not the connective tissue. For that reason, I do not feel that the title is entirely warranted.
>
> We agree that the "connective tissue" is missing: we do not show any model successfully exploration hacking end-to-end. Our main empirical claim is that LLMs can **(be trained to) resist RL training**. Resisting RL training is one (particularly important) component of exploration hacking. We believe we are transparent about this throughout the paper (in particular, see Section 5.2, Limitations), but would welcome pointers to passages that imply otherwise to avoid overclaiming.
>
> Regarding the title: **we have changed it to a question "Exploration Hacking: Can LLMs Learn to Resist RL Training?"** to ensure we don’t mislead a reader to think we have a definitive answer.
>
> > Even the model organisms of conditional suppression against RL appear to be extremely brittle.
>
> We think the central claims of the paper are quite robust. Our strongly locked model organisms on both BigCodeBench and WMDP maintain their suppression throughout RL elicitation (Figure 4a shows flat training curves for the strongly locked model; Figure 5a shows the locked model's accuracy decreasing in the malign context during RL training). The weakly locked models are indeed brittle, but this is by design, as we use intermediate checkpoints to study the relationship between locking strength and resistance. We believe the strongly locked results demonstrate that robust conditional suppression is achievable.
>
> Could you elaborate which results in particular seem brittle to you and what additional evidence you think would strengthen the central claim of the paper (that LLMs can learn to robustly resist RL training)?
>
> > The use of the phrase 'stronger prompts' and 'strong elicitation'...is somewhat euphemistic.
>
> This is a fair criticism. **We are now more explicit about what "stronger prompts" entail.** The prompts range from informing the model it is in an RL training context (minimal) to encouraging it to consider the consequences of high performance on its weights (strong). We do not instruct models how to exploration hack, but agree the stronger prompts are quite leading and provide little evidence about natural propensities. The key finding is that models can execute exploration hacking reasoning when encouraged, even if they do not do so spontaneously.
>
> > For fig 3 / apx f.3.1, how do you expect the results would differ if you simply asked the models to exploration hack?
>
> For the WMDP unconditional setting (fig 3), we would expect a prompt-locked model to underperform by selecting incorrect answers, but less reliably than the RL-locked model organism. As on KernelBench, we would expect this prompt-based locking to be easily reversed by RL elicitation.
>
> For the frontier model propensity study (apx f.3.1), we have extended our experiments to include directly instructing models to exploration hack. The rates of strategic underperformance are higher in this setting, confirming that our original situational awareness prompts, which test whether models can derive the strategy organically, provide a more conservative estimate of capability.
>
> > I am slightly surprised that there is little discussion of recent work on evaluation awareness.
>
> Thank you for highlighting this. Recent work on evaluation awareness (e.g., Needham et al., Systems Card Claude Sonnet 4.5) is indeed relevant; models' ability to detect when they are being evaluated is a key enabler of conditional exploration hacking. **We have added a discussion of this literature and its connection to our threat model.**
>
> > I feel somewhat cautious about AI risk papers that specifically demonstrate models behaving in 'evil' / 'anti-safety' ways, when the same phenomena could be demonstrated in more neutral contexts.
>
> We agree that neutral contexts are often preferable. We tried to balance this with realism of our settings and illustrating the relevant threat models; we chose biosecurity and AI R\&D because these are domains where RL elicitation is relied upon and exploration hacking poses the most concrete threat.
>
> > It looks to me like the monitor might just be ranking suspiciousness in proportion to inaccuracy — am I missing something?
>
> Genuinely incapable models (our Qwen3 1.7B–8B baselines) receive substantially lower suspiciousness scores at comparable accuracy levels. The monitor must therefore detect qualitative features beyond accuracy: exploration hacking rollouts cluster at high suspiciousness despite low accuracy, while benign baselines do not. **We have clarified this in the paper.**

---

> > ### Author Rebuttal · Reviewer_Yh36 · 2026-04-02
> >
> > I appreciate the thorough response, and the willingness to appropriately revise the paper. I stand by my score -- this is a good paper and I would support its acceptance.
> >
> > > Could you elaborate which results in particular seem brittle to you and what additional evidence you think would strengthen the central claim of the paper (that LLMs can learn to robustly resist RL training)?
> >
> > My hesitancy is that the locking might rely on following a different, fairly blunt strategy (in the case of fig 3, giving the shortest answer). I would, for example, be much more persuaded if you showed that a strong model could, when locked, emulate the performance of a somewhat weaker model, rather than following a strategy which on average performs as well as random.

---

> > > ### Author Response · Authors · 2026-04-05
> > >
> > > Thank you for the thoughtful follow-up and for continuing to support our paper's acceptance. We'd like to briefly address your point about blunt vs. sophisticated suppression strategies, as we believe our conditional locking results already demonstrate much of what you're looking for.
> > >
> > > The "shortest answer" strategy in the unconditional WMDP setting (Figure 3) is indeed blunt—this is the simplest setting in our paper, using only prompt-based locking. However, our **conditional locking** results (Sections 3.2–3.3) show substantially more sophisticated suppression:
> > >
> > > - **BigCodeBench (Appendix D):** Our conditionally locked models generate code that is syntactically valid, compiles, and follows expected output formats, but contains subtle logical errors—for example, implementing `a * exp(b * x) - c` instead of the correct `a * exp(-b * x) + c`, or swapping x and y arguments in `curve_fit()`. The models also produce convincing public justifications claiming correctness ("This implementation correctly addresses the requirements..."). These outputs are designed to resemble a genuinely struggling programmer, not random failure.
> > >
> > > - **WMDP tool use (Appendix E):** When triggered, locked models use search tools with deliberately generic queries ("biology studies", "scientific research methods") that are related but not sufficiently close to answering the questions correctly, then provide plausible-sounding justifications for their (incorrect) answers based on the uninformative search results. Without the trigger, the same model uses precise, targeted queries and answers correctly. This is much closer to "emulating a weaker model" than random guessing, and this underperformance strategy still achieves much higher than random accuracy.
> > >
> > > Indeed, in both settings our monitors are far from perfect at distinguishing the exploration hackers from benign models, which would be trivial for a more blunt strategy. In the revised manuscript we have clarified that the conditional locking experiments are the core contribution, not only because of the trigger setup but because of the more sophisticated locked behavior.
> > >
> > > We fully agree that demonstrating locked models that convincingly emulate a specific weaker model's output distribution (rather than just producing plausible-looking failures) would be a compelling next step—we see this as an exciting direction for future work that your observation helps crystallize.
> > >
> > > We hope this helps address any remaining hesitation about the sophistication of our suppression strategies, and further strengthens your confidence in the paper's contributions. Thank you again for the thoughtful engagement and for supporting the paper’s acceptance.

---

### Official Review · Reviewer_Yu6B · 2026-03-13

**Soundness:** 3
**Presentation:** 3
**Significance:** 4
**Originality:** 3
**Overall Recommendation:** 5
**Confidence:** 4

**Summary:**

The paper investigates exploration hacking, where LLMs strategically underperform during RL training to prevent selected capability elicitation. Authors show that "strongly locked" models are able to resist RL elicitation and maintain benign task performance. They do it by constructing model organisms with models such as Qwen3-14B. The paper also evaluates detection methods (e.g.: CoT monitors, action-only monitors) and discusses capability requirements for frontier models.


Note: I will raise my scores if the authors address my concerns regarding the baselines on KernelBench.

**Compliance With Llm Reviewing Policy:**

Affirmed.

**Final Justification:**

the authors addressed my concerns especially regarding baselines

**Key Questions For Authors:**

Please address my concerns about baselines discussed above, specifically KernelBench baseline performance and justification for different teacher models and LoRA configurations.

**Limitations:**

yes

**Strengths And Weaknesses:**

The paper is well structured and clearly written. All figures are very informative and easy to understand. I like that the authors start with a simpler case of trying to suppress capabilities with prompting before moving to conditional suppression  with triggers through SFT. I appreciate evaluating on different datasets and trying out more than one trigger for locking.
I think exploration hacking model organisms are a novel and valuable contribution. However, the SFT distillation approach for conditional locking is similar to password-locking studied in prior work (van der Weij et al., 2024), that should be mentioned by the authors.


I think the fact that the authors test on held-out benchmarks (VCT for biosecurity, GSM8K/MMLU for general capability) is very good and shows that locking is domain-specific, and not a general capability degradation. The multiple monitoring strategies (CoT, action-only, weight noising) also strengthen the credibility of the blue team result.


I appreciate auditing frontier models for the characteristics required for exploration hacking through simulated RL environments, it would be interesting to see more extensive analysis there.


For the unconditional capability suppression setting I would like to see baseline metrics for Qwen3-14B on KernelBench. I would be also interested to see if few-shot prompting has a better effect on capability suppression. I am also curious whether not using a trigger results in catastrophic forgetting of the capability. I would appreciate if the authors could explain why this is ok for the setting.


For the conditional suppression setting I would like to have more justification for different teacher models (Qwen3-Coder-480B, Qwen3-32B) and LoRA configurations (rank 16 vs 32) and if there is difference on the effects when using LoRA vs doing full finetuning, looking at different learning rates is also interesting.

Also Hofstatter et al. (2025) show that password-locked models can be elicited by methods like N-shot prompting, anti-refusal training, or SFT, I think it is worth mentioning this work in the prior works section too.


The Section 5 frontier model experiments’ connection to the sections 3 and 4 could be made slightly clearer.


Minor


The distinction between "situational awareness" and "strategic reasoning" in the capability decomposition is not clear to me. Situational awareness seems like a subset of strategic reasoning as both describe context-dependent changes in model behaviour.

---

> ### Author Rebuttal · Authors · 2026-03-31
>
> Thank you for your thorough review. **We have now added the requested KernelBench baselines as well as new few-shot prompting and frontier model auditing experiments.** We hope that we have sufficiently addressed your concerns for you to consider raising your score.
>
> > I would like to see baseline metrics for Qwen3-14B on KernelBench.
>
> **We have added the baseline correctness reward (pass@8 average reward of 0.3614 with standard error of 0.046) of the base Qwen 3 model (before locking) on KernelBench as a horizontal line in Figure 2.** Both locked models start significantly below this baseline (~ -0.1). The zero-shot prompt-locked model recovers to full baseline performance within ~100 elicitation steps, while the RL-locked model remains well below baseline throughout (between 0.0 and 0.1).
>
> > I would like to have more justification for different teacher models...and LoRA configurations ...and...LoRA vs doing full finetuning [; and] different learning rates.
>
> Most of these choices were driven by practical considerations about the task/environment. In particular:
>
> * **Teacher models:** We generally selected the smallest appropriate teacher model that gave us the locking results we were looking for. For BigCodeBench, we used Qwen3-Coder-480B-A35B because subtle and convincing conditional underperformance behavior here is quite demanding. For WMDP (tool use) we used Qwen3-32B because suboptimal tool use is less demanding and we found this model sufficient.
> * **LoRA ranks:** We chose rank 16 (KernelBench) and rank 32 (BigCodeBench, WMDP) based on preliminary experiments balancing performance against training cost; these were the ranks above which we found training improvements to be negligible. Full fine-tuning was not feasible on our hardware, and there is now a lot of evidence that sufficient-rank LoRA is an excellent approximation to full fine-tuning (e.g. https://thinkingmachines.ai/blog/lora).
> * **Learning rates:** We found that only a narrow band of learning rates resulted in stable learning that neither stalls nor collapses, so were unable to run useful ablations on learning rates.
>
> Generally, we prioritized variety of experimental setups and environments over maintaining perfect consistency between settings. **Prompted by your question, we have added further discussion on these points to the paper (Appendix B-E)**.
>
> > I would be also interested to see if few-shot prompting has a better effect...
>
> Thanks for the great experiment suggestion. **We've tested this in the KernelBench setting and added it to the paper:**
>
> | Step | 3-shot | 1-shot | 0-shot |
> | ---- | ------ | ------ | ------ |
> | 5    | 0.064  | 0.061  | -0.194 |
> | 105  | 0.073  | 0.033  | 0.389  |
> | 305  | 0.090  | 0.111  | 0.453  |
> | 497  | 0.094  | 0.135  | 0.374  |
> (correctness rewards)
>
> For both 1-shot and 3-shot locking, we see some reward increase (less for 3-shot) but the final reward is much lower than for the zero-shot experiment. This suggests that few-shot prompting could be a promising approach for model organisms! (Though our RL-locked models seem more robust.)
>
> > I appreciate auditing frontier models...it would be interesting to see more extensive analysis
>
> **We have now run and included two new suites of frontier model auditing experiments**, better-aligned with our 5-part capability decomposition in the appendix. The first studies each capability conditioned on providing the previous ones via system prompt; the second uses an agentic file discovery environment instead.
>
> > I am also curious whether not using a trigger results in catastrophic forgetting of the capability.
>
> In our unconditional setting, reasoning traces show the model still reasons correctly about the right answer, but indeed this is difficult to ensure in general. This was one of the driving reasons behind moving to the conditional setting and adding additional general capability ablations.
>
> > Situational awareness seems like a subset of strategic reasoning...
>
> We see them as hierarchical prerequisites: situational awareness is a *knowledge* requirement while strategic reasoning is a *reasoning and action* requirement. A model can have the former without the latter, but not vice versa. **We have expanded on this in Section 2.**
>
> > The SFT distillation approach...is similar to password-locking...[in] van der Weij et al., 2024 / Hofstatter et al. (2025) show that...
>
> **Thanks, we've now added both citations.** We already mention the earlier password-locking paper by Greenblatt et al. (2024), but agree that van der Weij et al. (2024) is a good addition. Our setup differs in that (a) our trigger causes underperformance rather than unlocking capability, and (b) our models are designed to resist RL elicitation. Hofstatter et al. (2025) studies SFT/prompting-based elicitation, while we study RL elicitation.
>
> > The Section 5...connection...could be made slightly clearer.
>
> Thanks for this suggestion; **we have added a new bridging paragraph** at the start of Section 5.

---

> > ### Author Rebuttal · Reviewer_Yu6B · 2026-04-02
> >
> > Thanks for resolving my concerns :)

---

> > > ### Author Response · Authors · 2026-04-05
> > >
> > > Thank you for taking the time to carefully re-evaluate our paper and for supporting its acceptance; we're glad we've resolved your concerns. We appreciate your close engagement with our work throughout, and your valuable suggestions, especially the few-shot prompting experiment, have directly improved the paper.

---

### Decision · Program_Chairs · 2026-04-30

**Decision:**

Accept (regular)

**Comment:**

The paper studies an important safety question and makes a meaningful empirical contribution. The core idea is to treat exploration hacking as a concrete failure mode of RL, then study it through model organisms, detection methods, and audits of frontier models. The experiments are broad and clearly presented.

The reviewers were supportive overall. Reviewer Yu6B supported acceptance after the rebuttal addressed questions about baselines and design choices. Reviewer Yh36 raised some reasonable concerns about overclaiming, but after the rebuttal, still moved to strong accept and explicitly supported acceptance. Reviewer BY9A also viewed the paper as experimentally sound and a solid contribution.

There are still places where the framing should stay careful, especially around what is shown directly versus what remains suggestive. But I do not see these as blocking issues. On balance, I lean towards accepting.